# Adopting Graph Neural Networks to Analyze Human–Object Interactions for Inferring Activities of Daily Living

**DOI:** 10.3390/s24082567

**Published:** 2024-04-17

**Authors:** Peng Su, Dejiu Chen

**Affiliations:** Department of Engineering Design, KTH Royal Institute of Technology, 100 44 Stockholm, Sweden; pensu@kth.se

**Keywords:** graph neural network, scene understanding, activities of daily living analysis

## Abstract

Human Activity Recognition (HAR) refers to a field that aims to identify human activities by adopting multiple techniques. In this field, different applications, such as smart homes and assistive robots, are introduced to support individuals in their Activities of Daily Living (ADL) by analyzing data collected from various sensors. Apart from wearable sensors, the adoption of camera frames to analyze and classify ADL has emerged as a promising trend for achieving the identification and classification of ADL. To accomplish this, the existing approaches typically rely on object classification with pose estimation using the image frames collected from cameras. Given the existence of inherent correlations between human–object interactions and ADL, further efforts are often needed to leverage these correlations for more effective and well justified decisions. To this end, this work proposes a framework where Graph Neural Networks (GNN) are adopted to explicitly analyze human–object interactions for more effectively recognizing daily activities. By automatically encoding the correlations among various interactions detected through some collected relational data, the framework infers the existence of different activities alongside their corresponding environmental objects. As a case study, we use the Toyota Smart Home dataset to evaluate the proposed framework. Compared with conventional feed-forward neural networks, the results demonstrate significantly superior performance in identifying ADL, allowing for the classification of different daily activities with an accuracy of 0.88. Furthermore, the incorporation of encoded information from relational data enhances object-inference performance compared to the GNN without joint prediction, increasing accuracy from 0.71 to 0.77.

## 1. Introduction

Human Activity Recognition (HAR) involves multiple techniques to analyze sensory data [1]. These sensory data constitute a basis for assessing and predicting human activities. In the field of Human Activity Recognition (HAR), the applications of smart homes and assistive robotic systems are paving the way to support individuals in performing their Activities of Daily Living (ADL), therefore facilitating and monitoring their quality of life [2]. Various equipment collect operational conditions and human status by employing wearable sensors like wrist-worn accelerometers [3] and non-wearable sensors like cameras [2,4] to attain the recognition of ADL. Compared to wearable sensors, the adoption of camera frames to analyze and classify Activities of Daily Living (ADL) presents a promising solution due to the inherently multifarious features found in image data [5,6,7]. Most of the approaches utilize image frames to detect ADL by combining pose estimation with skeleton-based action recognition [8,9]. Methods based on Convolutional Neural Networks (CNN) typically demand significant effort to identify key points and joints of human bodies. As shown by [10,11], complex human motion capture systems can be used to support annotating the key points through extensive data. With such data, a variety of CNN architectures can be trained to estimate pose by formulating body joints and extracting features [12,13]. Many Graph Neural Networks (GNN)-based solutions have been considered to be support for alleviating the need for deep architectures to extract the features from the images, as such solutions capture the key points and joints with graph models [14,15,16,17]. Through the analysis of graph models representing skeleton-based human bodies, GNN can be used to estimate the likelihood of human actions. However, the uncertainties stemming from the probabilistic nature of neural networks [18,19] often necessitate extensive training data with high sensory resolution for accurately identifying the human body parts [8,9,13,14,16]. These requirements restrict the applicability of cameras for recognizing daily activities in the context of assisting at-home scenarios.

To address this issue, we propose a framework where GNN are adopted to explicitly analyze human–object interactions for inferring human activities of daily living alongside the corresponding environmental objects. Specifically, the framework first extracts the relational data on the interactions between humans and environmental objects from the collected image frames. Next, GNN automatically encodes the correlations among the interactions indicated by the respective relational data and, therefore, detects the presence of activities and their environmental objects, leading to a more effective analysis of ADL. We present the contribution of this paper as follows:Designing a conceptual framework to construct graph-based data by image frames to infer the ADL within assisting at-home applications.Proposing a GNN architecture to jointly predict environmental objects and ADL by comprehending the relational data.Enhancing the prediction accuracy of ADL and environmental objects by aggregating the encoded information from the semantics of relational data.

The rest of the paper is organized as follows: Section 2 presents prior work related to GNN with environmental scene understanding. Section 3 describes the proposed framework. Section 4 presents a case study by verifying the proposed framework with the Toyota Smart Home dataset. Section 5 presents the conclusion of the proposed framework and discusses the future work.

## 2. Related Work

This section first provides background information on GNN. Next, we present previous work on GNN applied in the applications related to the topic. In addition, we exhibit current efforts to apply image frames to relational data in scene understanding.

### 2.1. Background of GNN

GNN are specifically designed for processing non-Euclidean data, supporting the analysis of graph-based data [20]. Such graph-based data structures usually consist of nodes and edges to represent a set of objects and relations. Specifically, graphs can be classified into heterogeneous graphs, which typically connect nodes with different types of edges, and homogeneous graphs, where edges do not convey additional information [20]. A variety of GNN models are used to analyze these two graphs regarding their spatial and temporal properties [21]. Spatial models support the transformation of graph-based data into a spectrum space using Graph Laplacian [22,23] or encoding information from local neighbors of specific nodes through aggregation operations [24] with Graph Convolutional Networks (GCN). Building on the spatial models, the adoption of gate mechanisms from RNN and LSTM is a common solution to enable temporal analysis of graph-based data [21].

### 2.2. GNN to Cope with HAR and ADL

Most GNN integrate different models to analyze human activities by synthesizing spatial–temporal features. As mentioned earlier, some of them recognize the key points of the human body by analyzing unstructured high-dimensional data such as video clips [9,25]. These high-dimensional data could either contain video clips with depth information as 3D data or solely rely on raw 2D images captured by cameras [26,27]. Depending on the input data formats, these GNN can be roughly categorized into the following trends [9]: (1) Spatio-temporal GCNs encode the key points of human bodies as nodes in graphs, while the evolution of human activities is usually interpreted as attributes of edges among the nodes within the graphs [28]. This method usually requires the analysis of the graphs, including all elements, such as edges and nodes, to identify human activities. However, to accurately identify the key points of human bodies, such a method usually requires high-resolution data or additional depth information. As an example in [29], the input data requires annotating bones and joints within human bodies with depth information, which decreases the generalization of the proposed framework. (2) Temporal-aware GCNs focus on extracting contextual dependencies in sequential data by adopting and optimizing attention mechanisms. This method typically analyzes contextual information across video sequences with similar lengths. However, due to the diversity of activities within video sequences, attention-based methods could become more time-consuming and less efficient [30,31]. (3) Multi-stream GCN refers to an integration with different inputs for identifying human activities. A typical example in [15,17] usually uses video clips and skeleton-based data as two-stream input for GCN to extract features. This method aims to identify human daily activities by aggregating image frames and incomplete skeleton-based data, reducing the reliance on high-resolution and well-annotated datasets. While these methods enhance the efficiency of detecting human activities, further efforts are needed to understand the interaction between humans and environmental objects. Towards this direction, we also investigate previous work on scene understanding through the utilization of GNN.

### 2.3. Applying Relational Data to Scene Understanding

One common solution is to adopt GNN to analyze and understand scenes in image frames. Such GNN support inferring common-sense relationships among objects within scenes [32,33,34]. Therefore, a critical step in utilizing GNN for scene understanding is to convert high-dimensional unstructured data (e.g., image frames) into relational context within a graph-based structure. A basic process for constructing such graph-based data is to extract objects within image frames as nodes. The edges between nodes represent pairwise relations between the objects, depicting their spatial and temporal evolution. The semantics of graph-based data are analyzed through the adoption of GCN. However, this implementation could be insufficient for understanding the task-specific scene. For example, when a human detected to be overlapping with a motorcycle is represented in graph-based data and analyzed by the GCN, their relationship is highly likely to be recognized as the human riding the motorcycle in a public area. However, when this human is riding a motorcycle without a helmet, these methods may not capture insights into unsafe behaviors. Hence, combining task-specific scene understanding with certain prior knowledge aids in achieving specific tasks. The presentation of such prior knowledge could be categorized as follows: (1) Explicit rules refer to directly leveraging human knowledge imposed into the graph-based data. In [35,36,37], objects from Bird’s-Eye Views (BEV) within dynamic driving scenarios are converted into graph-based data to facilitate analysis by GCN, incorporating specific traffic rules and common-sense knowledge. A typical human-understandable rule is exemplified in [36], where the weighted edge within the node represents the relative distance. GCN are used to analyze potential node pairs whose relative distance violates specified rules. However, these methods usually require landmarks (e.g., static objects) to annotate the relationships among objects, which limits their generalization for extension in ADL-related applications. (2) Encoded formal knowledge refers to the process of interpreting human knowledge into machine-readable specifications. For example, in [38], common-sense knowledge is converted into propositional logic to be incorporated with GCN in the context of recommendation systems.

Inspired by the aforementioned methods of understanding scenes, we introduce a GNN-based framework designed to comprehend scenarios within ADL-related applications. Unlike the conventional approach of relying solely on pose estimation for daily activities prediction [15,17], our proposed method achieves joint prediction by mapping the interactions, alleviating the need for skeleton-based data as part of the input. Compared with existing methods adopted in [35,36,37], the proposed method interprets common-sense knowledge into temporal logic specifications without relying on landmarks for further annotating the relationships.

## 3. Methodology

In this section, we present the framework shown in Figure 1 to infer activities of daily living. We describe the main workflow of the proposed work as follows:

### 3.1. Relational Data Construction

We construct relational data for GNN analysis by extracting interactions from image frames. Specifically, the relational data in terms of graph-based data consists of nodes and edges. The objects in the video clips are extracted as nodes in the graph models, while the interactions within these objects are represented as edges. Therefore, the following steps outline the process to obtain these graph models:

#### 3.1.1. Node Extraction

At this step, we obtain the node information required for creating graph-based data. We define the nodes based on the information presented in image frames. Specifically, we formulate $D^{a_{i}}$ for a video clip collected from a scenario $a_{i}$ as follows:(1)$$D^{a_{i}}=\left\{d_{1}^{a_{i}},d_{2}^{a_{i}},\cdots ,d_{n}^{a_{i}}\right\}\;$$
where *n* refers to the number of frames in the video clip $D^{a_{i}}$. $a_{i}$ refers to a specific daily activity obtaining a label $y_{a}\in T_{a}$. $T_{a}$ represents a set of labels for daily activities collected in the dataset.

An object-detection module $M_{n}\left(\cdot \right)$ is used to identify the nodes of the graph model by extracting the objects in any frames $d_{k}^{a_{i}}$ of $D^{a_{i}}$. We formulate the process as follows:(2)$$O_{k}^{a_{i}}=M_{n}\left(d_{k}^{a_{i}}\right)$$
where $O_{k}^{a_{i}}=\left\{o_{1}^{k},o_{2}^{k},\cdots ,o_{j}^{k}\right\}$ refers to the collection containing the objects extracted from the video clips. Each $o_{i}^{k}$ from $O_{k}^{a_{i}}$ is a vector denoting the features of an object, such as its bounding box sizes and object types. Each detected object $o_{i}^{k}$ obtains a label $y_{o}\in T_{o}$ indicating the types of object. $T_{o}$ represents a set of labels for environmental objects collected in the dataset.

#### 3.1.2. Edge Extraction

To represent the relationships within the video clip $D^{a_{i}}$, it is critical to analyze the spatial and temporal properties of human and environmental objects. We label these relationships via the edges across nodes. As mentioned earlier, existing studies typically employ data-driven approaches, such as LSTM, to extract relationships by encoding input features from extensive graphs [35,36,37]. However, the duration periods within different daily activities could exhibit extreme variety [17]. For example, drinking water in the kitchen could be captured in a few image frames, while recognizing activities like washing dishes in the same place may require more images. Therefore, using data-driven methods could be inefficient for encoding an entire video clip. In contrast, the knowledge could enhance the efficiency of data-driven methods in task-specific scenarios (e.g., human action reasoning [16,32] and recommendation systems [39]) that involve possible known relationships. Since activities of daily living typically involve well-known interactions between humans and environmental objects, we propose a rule-based method for extracting the relationships of nodes. Similar rule-based methods also can be found in [16,37]. Specifically, we formulate the rule to identify the interactions by temporal logic specifications:(3)⋄(ϕ∪(T∧(¬ϕ∪ρ)))
where *T* refers to the time duration, and $\rho ::=(Occ_{m_{ij}}\geq n)$, where $Occ_{m_{ij}}$ refers to the number of appearances in the video clip $D^{a_{i}}$, *n* refers to the threshold of occurrence number. $\phi ::=(m_{ij}\geq \tau$) denotes the condition when the interaction rate $m_{ij}$ for objects $o_{i}^{k},o_{j}^{k}$ in a single frame *k* exceeds a threshold τ.

We formulate the interaction rate $m_{ij}$ as Equation (Equation 4), which is identified by the Intersection over Union (IoU) areas between a pair of objects with non-maximal suppression [32,33].
(4)$$m_{ij}=\frac{I(xy_{i}^{k},xy_{j}^{k})}{U(xy_{i}^{k},xy_{j}^{k})}$$
where $xy_{i}^{k},xy_{j}^{k}$ refer to the bounding box sizes of $o_{i}^{k},o_{j}^{k}$. These sizes are obtained by the object-detection module $M_{n}\left(\cdot \right)$. $I(xy_{i}^{k},xy_{j}^{k})$ refers to the intersection area within the objects, while $U(xy_{i}^{k},xy_{j}^{k})$ refers to the union area within the objects. Once $m_{ij}$ satisfy the rule defined by Equation (Equation 3), we denote the interaction as $<o_{i}^{a_{i}},r_{i,j}^{a_{i}},o_{j}^{a_{i}}>$, where $r_{ij}\in M^{a_{i}}$. $M^{a_{i}}$ denotes a set of identified interactions within detected objects from the video clip $D^{a_{i}}$. Furthermore, we denote all interaction pairs in the context of a graph $G^{a_{i}}$ as follows [24]:(5)$$G^{a_{i}}=\left\{\left(o_{i}^{a_{i}},r_{i,j}^{a_{i}},o_{j}^{a_{i}}\right)\right\}$$

Additionally, each generated graph $G^{a_{i}}$ obtains a label $y_{a}\in T_{a}$ indicating the type of daily activities. $T_{a}$ refers to a set of labels for the daily activities.

### 3.2. Joint Prediction via GNN

After the relational data construction phase, we utilize Message-Passing Neural Networks (MPNN) [40] to integrate GNN models for the joint prediction (see Figure 1).

#### 3.2.1. Message-Passing Phase

This step involves the computation for aggregating and updating information from the neighbors of a specific node along with the edges of shared relationships. Specifically, we model the message aggregating process in the layer *l* as follows:(6)$$m_{i}^{l+1}=\sum_{j\in N(i)}M_{l}\left(h_{i}^{l},h_{j}^{l},r_{i,j}\right)$$
where i,j are the same as Equation (Equation 5), $r_{i,j}\in t_{i,j}^{a_{i}}$ refers to the edge types connecting from $o_{i}^{a_{i}}$ to $o_{j}^{a_{i}}$. We denote $h_{i}^{l}$, $h_{j}^{l}$ as the encoded information of the node $o_{i}^{a_{i}},o_{j}^{a_{i}}$ in layer *l*. This encoded information is dependent on the configuration of the message-passing network. As an example, $h_{i}^{l},h_{j}^{l}$ are equivalent to the features within $o_{i}^{a_{i}}$ and $o_{j}^{a_{i}}$, respectively, when l=1. N(i) refers to the set of all neighboring nodes of the node $o_{i}^{a_{i}}$ whose example is shown in Figure 1. $M_{l}\left(\cdot \right)$ refers to message-passing functions, such as concatenation and multiplication operations. Equation (Equation 6) shows that by computing all the neighboring nodes N(i) in terms of message passing, $m_{i}^{l+1}$ merges the information from the features of both the target node and their contextual nodes.

To further encode the aggregated relational data, the network propagates the edge information within the neighbors by creating an edge (vertex) updating function $U_{l}$ as follows:(7)$$h_{i}^{l+1}=U_{l}\left(h_{i}^{l},m_{i}^{l+1}\right)$$
where $U_{l}$ refers to a composition of non-linear functions, such as a ReLU function and recurrent units.

#### 3.2.2. Readout Phase

In this step, the readout operation approximates feature vectors z for the graph-based data $G^{a_{i}}$. We use multiple embedding $z\in \{z_{a}^{a_{i}},z_{o}^{a_{i}}\}$ to encode the information of activities ${\hat{y}}_{a}$ and environmental objects ${\hat{y}}_{o}$ within the context of the graph $G^{a_{i}}$. The embedding vectors z are formulated as follows:(8)$$z=R(\left\{h_{i}^{L}|i\in G^{a_{i}}\right\})$$
where $R\in \{R_{a},R_{o}\}$. $R_{a}$ and $R_{o}$ refer to readout functions, configurable with various operations, such as a linear layer and sum operation, to generate $z_{a}^{a_{i}}$ and $z_{o}^{a_{i}}$, respectively. *L* refers to the running steps in the message-passing phase.

Considering daily activities involving interactions between humans and objects, the predicted object classes are often correlated with these activities. For instance, eating in a kitchen is a typical daily activity commonly associated with specific environmental objects such as bowls [17]. However, detecting bowls in the kitchen is insufficient to confirm that humans are eating. Therefore, we propose an aggregation operation A(·) to enhance the performance of predicting environmental objects by synthesizing embeddings $z_{a}^{a_{i}}$ and $z_{o}^{a_{i}}$ as follows:(9)$$z_{c}^{a_{i}}=A\left(z_{a}^{a_{i}},z_{o}^{a_{i}}\right)$$
$z_{c}^{a_{i}}$ refers to an aggregated embedding to predict environmental objects. To this end, we model the output layers as follows:(10)$$\hat{y}=F\left(z_{e}\right)$$
where $z_{e}\in \left\{z_{c}^{a_{i}},z_{a}^{a_{i}}\right\}$, F(·) refers to the configuration of output functions to predict activities and objects, where $F\in \{F_{a},F_{o}\}$, $\hat{y}$ refers to the predicted results, where $\hat{y}\in \left\{{\hat{y}}_{a},{\hat{y}}_{o}\right\}$. ${\hat{y}}_{a}$ denotes the predicted activities of daily living using the output function $F_{a}$ with embedding $z_{a}^{a_{i}}$, while ${\hat{y}}_{o}$ represents the predicted classes of environmental objects using the output function $F_{o}$ with the aggregated embedding $z_{c}^{a_{i}}$.

## 4. Case Study

In this section, we elaborate on the implementation of the proposed framework. First, we provide a brief introduction along with an explanation for selecting the Toyota Smart Home dataset. Next, we present the configuration of relational data construction and joint prediction based on this dataset. Finally, we present the results in comparison with baseline methods.

### 4.1. Overview of Toyota Dataset

The Toyota Smart Home dataset [17] is a set of video clips collected from different locations of an apartment whose Bird Eye View (BEV) is shown in Figure 2. The reasons for selecting this dataset to evaluate the proposed methods are as follows: (1) It contains over 10,000 video clips captured from different locations in the apartment, providing diversity to record various daily activities. (2) The resolution of video clips captured by cameras is 640×480, challenging the identification of human body parts. In this case, understanding between humans and environmental objects provides a promising solution for detecting daily activities.

Specifically, we choose three camera views shown in Figure 2 recording from the dining room, living room, and kitchen. To evaluate the proposed framework, we particularly selected 8 daily activities, including eating meals, calling phones, and using laptops. These activities commonly involve the interaction between humans and environmental objects. To reduce the correlation between the daily activities and locations, these activities could occur in multiple locations. Additionally, we select video clips that feature multiple types of daily activities occurring in the same location, as well as the same activity taking place in different locations. For example, a person could use a cell phone in both locations shown in Figure 2 while also engaging in cooking and cleaning in the kitchen.

### 4.2. Constructing Relational Data

Existing deep neural networks designed for object detection could be employed in the object-detection module $M_{n}$. In this paper, we adopt a pre-trained Fast-RCNN model to detect objects in the video clips [41]. Every node consists of the types and bounding box sizes from the detected objects. Moreover, we assign an ID to each detected object to prevent duplicating the same types of objects occurring in the images. To extract relationships from the video clips, we set the IoU threshold to τ=0.4. If $m_{ij}$ exceeds the threshold until more than n=20 instances or appears continuously for more than T=0.2 length in the image sequences throughout the entire video clips, we annotate that the relationships between objects *i* and *j* are engaged in interaction. In particular, we annotate relationships between people and environmental objects when constructing graph-based relational data. Furthermore, we incorporate the location information of the video clips to enrich these data and facilitate the GNN in aggregating node features.

As a result, we extract 33 different types of environmental objects. The following daily activities are extracted from the dataset: cleaning, cooking, watching TV, eating food, reading books, using the telephone, using a laptop, and drinking water. Except for cleaning, cooking, and watching TV, the rest of the activities could occur in multiple locations. As illustrated in Figure 3, we present the graph-based relational data of daily activities extracted from various locations. From Figure 3, we note that even though the person is cleaning and cooking in the same location, the edges in the graph for these two daily activities still depict different connections. Specifically, when the person is cooking, there is more interaction between the person and the bowls and the refrigerator. In contrast, when the person is cleaning, the edges are more connected to the person, bottles, and sink. Moreover, the remaining activities also manifest significant features within the context of relational data. For instance, during eating, interactions typically occur with items such as tables, chairs, and dishes. Similarly, when watching TV, interactions involve remotes and humans.

### 4.3. Implementing Joint Prediction via GNN

We adopt two-layer message-passing networks whose layout is shown in Figure 1 to encode the information from input graphs. We use GraphSAGE in the first layer to attain encoding the features of the edges and nodes [40,42]. Specifically, We use a mean aggregator shown in Equation (Equation 11) as the message-passing function $M_{l}$.
(11)$$m_{i}^{l+1}=\underset{j\in N(i)}{⨁}\left(h_{i}^{l},h_{j}^{l},r_{i,j}\right)$$
where ⨁ refers to approximate element-wise mean value from the encoded information $h_{i},h_{j}$ with their edge type $r_{i,j}$.

We adopt graph convolutional operators (GCNConv) with Laplacian-based methods based on [23] to attain message-passing functions in the embedding layers. Specifically, we model the message function as follows:(12)$$m_{i}^{l+1}=D^{\frac{1}{2}}AD^{-\frac{1}{2}}h^{l}W^{l}$$
*D* refers to the degree matrix. *A* refers to the adjacency matrix. $W^{l}$ refers to layer-wise learnable parameters in the *l*-th layer [23,40].

This layer consists of two parallel GCNConv, which are used to separately generate the embedding $z_{a}^{a_{i}}$ and $z_{o}^{a_{i}}$ from a video clip $a_{i}$. We use tanh functions as the edge updating function $U_{l}$ in each layer. We propose an element-wise multiplication operation as A(·) to aggregate the correlated features within $z_{a}^{a_{i}}$, $z_{o}^{a_{i}}$ and generate $z_{c}^{a_{i}}$. To this end, we use SoftMax classifiers as the output layers to generate the likelihood of prediction results ${\hat{y}}_{a},{\hat{y}}_{o}$ from $z_{a}^{a_{i}},z_{c}^{a_{i}}$, respectively. Sequentially, we define the loss function L as follows:(13)$$L=L_{c}\left({\hat{y}}_{a},y_{a}\right)+L_{c}\left({\hat{y}}_{o},y_{o}\right)$$

We train the parameters in the network by optimizing the loss function L, where $L_{c}$ refers to the cross-entropy between the predicted results and the ground-truth label. To this end, we develop a GNN-based framework to classify the graph-based content ${\hat{y}}_{a}$ under-recognized nodes and edges and to predict nodes ${\hat{y}}_{o}$ within a given graph. This framework synthesizes human–object interaction to infer activities of daily living.

### 4.4. Ablation Study

The training platform is configured with an AMD Ryzen 7 5800 and NVIDIA RTX-3070. During the training of the proposed methods, we collect all these daily activities, with each activity containing 600 graphs. We configure the training ratio to 0.8, and the training epoch is 800. We select multiple baseline methods to evaluate the proposed method. Specifically, we employ an MLP with two hidden layers to infer activities and objects by solely analyzing the features of nodes. This MLP configuration is equivalent to concatenating the intermediate embeddings from Fast-RCNN in Equation (Equation 2) to dense layers. In addition, we introduce two GCN designs, GNN with Split Prediction (S-GNN) and Attention-based GNN (Att-GNN), to evaluate their performance using the same dataset as the comparison. S-GNN shares the same network topology in [28] to analyze spatial properties of the graph-based data. This S-GNN adopts graph convolutional and dense layers to concatenate the features within the nodes from graphs. Att-GNN identifies correlations by modeling an energy function and attention distributions within spatial and temporal properties, enabling the analysis of graph and node patterns. In our case, we implement a similar network architecture used in [37], wherein a self-attention layer is connected behind the graph convolutional layers by replacing the multiplication operation A(·). As an ablation test, we additionally construct a Joint-Prediction Network (JP-GNN) by removing the operation A(·) and directly predicting the data.

The final results are shown in Table 1, where we conclude that the proposed method demonstrates significantly superior performance compared with MLP. Such results indicate that the relationships within the nodes empower the capability to infer daily activities and objects. Unlike GNN-based approaches, the inference process of MLP does not explicitly incorporate semantic context within graphs, owing to the inherent properties of feed-forward networks. Among GNN-based approaches trained for the same number of epochs, our proposed method achieves higher accuracy compared to the attention-based method, which also analyzes correlations within the embeddings. The possible reason for this situation could be that the attention-based method requires more time to attain convergence in the attention mechanism (e.g., learnable parameters in score functions). Compared with the JP-GNN which does not include the aggregation function, our proposed method shows significant improvement in object inference. These results indicate that the activity classification embedding aids in inferring objects. Additionally, the embeddings of activities and objects share the same layer, therefore affecting the convergence of the network. As a result, the TOP-1 accuracy of activities classification of JP-GNN is lower than that of our methods and the S-GNN which infers objects and activities separately. We also observe that the TOP-1 accuracy of activity inference from the proposed method is slightly higher than those of S-GNN. We believe that the reason could be the implementation of multiple embeddings serving as regularization to optimize networks. Similar situations also could be observed in prior studies, such as [22,39]. To further evaluate the performance of the proposed method, we also utilize the F1-score in Equation (Equation 14) by leveraging the Confusion Matrix in multi-classification cases [43,44].
(14)$$\begin{array}{cc} \; & Pr@y_{a}^{k}=\frac{TP@y_{a}^{k}}{TP@y_{a}^{k}+FP@y_{a}^{k}}\\ \; & Re@y_{a}^{k}=\frac{TP@y_{a}^{k}}{TP@y_{a}^{k}+FN@y_{a}^{k}}\\ \; & F1@y_{a}^{k}=\frac{2\times Re@y_{a}^{k}\times Pr@y_{a}^{k}}{Re@y_{a}^{k}+Pr@y_{a}^{k}} \end{array}$$
where $TP@y_{a}^{k},TN@y_{a}^{k},FP@y_{a}^{k},FN@y_{a}^{k}$ refer to True Positive, True Negative, False Positive, and False Negative in the Confusion Matrix. $Pr@y_{a}^{k},Re@y_{a}^{k},F1@y_{a}^{k}$ refer to the precision, recall and F1-Score at any activities $y_{a}$ with label *k*. Table 2 presents the overall results with Equation (Equation 14). Compared to the other activities, the proposed method shows poorer performance in identifying cooking and cleaning. This situation could be implied by the presence of common interacting objects in these two activities. For instance, both cooking and cleaning involve bowls and dishes in the same location. Additionally, the location of the camera in the kitchen, as shown in Figure 2, may introduce some uncertainty in efficiently detecting interactions between cookstoves and humans during cooking. This situation could be improved by utilizing image frames from multiple camera views with different locations.

Additionally, we evaluate the time consumption of training each method. With the same hyper-parameters (e.g., training epoch, batch sizes), S-GNN takes approximately 9 and 33 min to train the network to attain stable performance, respectively. Att-GNN requires more than 25 min to train the joint prediction. The proposed method takes around 21 min. These results indicate that compared with baseline methods, the proposed method spends less time to attain better performance.

## 5. Discussion and Future Work

This paper presents a framework to jointly infer the daily activities and environmental objects. Specifically, compared to the baseline methods, our framework demonstrates competitive performance in terms of TOP-1 accuracy and training efficiency. The proposed method supports incorporating semantic content within relational data rather than directly relying on high-dimensional data. This approach offers an explicit solution for inferring human daily activities and environmental objects. Compared to prior work on GCN related to the identification of human daily activities, the proposed method avoids the need for skeleton-based data and reduces reliance on complex training data. However, the proposed work relies on the semantics in the context of interaction between humans and the environment to identify the objects and daily activities. Such a mechanism could be inefficient in specific scenarios (e.g., entering and leaving).

Therefore, the following aspects could be future works: (1) Combining knowledge-aware approaches (e.g., knowledge graphs) with embedding to enhance the explainability and performance of the proposed networks. In contrast to the temporal logic constraints imposed in the proposed framework, domain knowledge can be encoded within the GCN-based framework to offer flexible constraints. (2) Utilizing recurrent units (e.g., LSTM) to reduce the labeling data and improve the generalization by encoding the temporal evolution. The proposed method can integrate various embeddings to encode and analyze temporal correlations. This encoded evolution is expected to enhance the granularity of daily activities, enabling the decomposition of activities (e.g., entering can be decomposed into opening doors and walking). (3) Extending the proposed framework to diverse datasets with complicated scenarios such as dynamic driving scenarios. In such scenarios, environmental objects exhibit various correlated behaviors, posing challenges in modeling and analyzing relational data in terms of their relationships and types. An extension of the proposed work targeting heterogeneous graphs with weighted edges could address these scenarios.

## Figures and Tables

**Figure 1 sensors-24-02567-f001:**
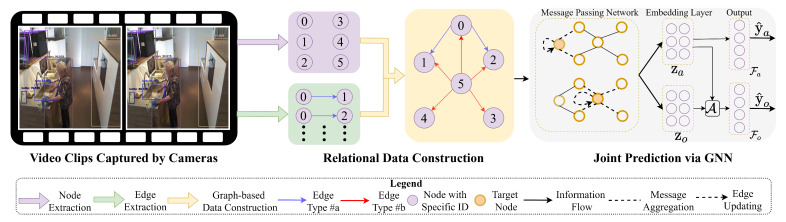
An example to present the overall process of extracting and constructing relational data. The edge types #a and #b refer to interactions with different features extracted from temporal specifications, as defined by Equation (Equation 3).

**Figure 2 sensors-24-02567-f002:**
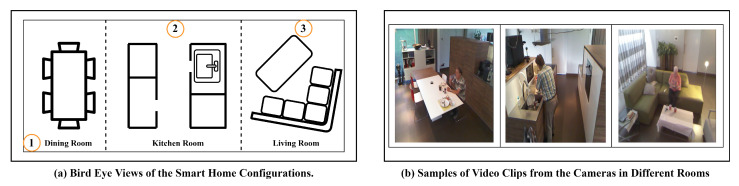
Bird Eye View (BEV) of the apartment. The numbers in the figure refer to the location of the camera installation. ➀, ➁, and ➂ refer to the camera locations used to capture video clips of activities.

**Figure 3 sensors-24-02567-f003:**
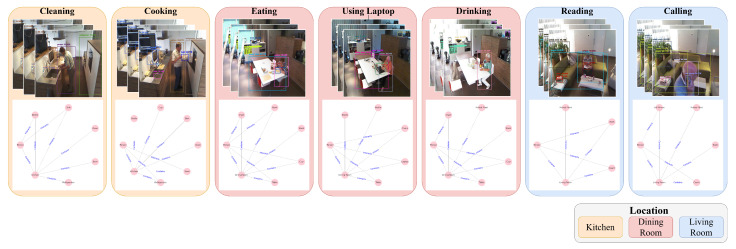
Samples of graph-based relational data generation based on the image frames.

**Table 1 sensors-24-02567-t001:** TOP-1 Accuracy of Different Methods.

	MLP	GNN-Based Methods			
		Our Method	Att-GNN	JP-GNN	S-GNN
Activities Inference	0.49	**0.88**	0.82	0.83	0.86
Objects Inference	0.56	**0.77**	0.65	0.71	0.68

**Table 2 sensors-24-02567-t002:** Precision, Recall and F1-Score Comparison.

	Reading	Cooking	Cleaning	Eating	Drinking	Using Laptop	Calling	Wathcing TV	Average
Precision	0.94	0.71	0.75	0.89	0.78	0.95	0.90	0.92	0.86
Recall	0.66	0.63	0.67	0.72	0.84	0.91	0.91	0.83	0.77
F1-Score	0.77	0.67	0.71	0.85	0.81	0.93	0.90	0.87	0.81

## Data Availability

Data available on request from the authors.

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
