# Peer review of "Adopting Graph Neural Networks to Analyze Human–Object Interactions for Inferring Activities of Daily Living"

_sensors, 2024, doi:10.3390/s24082567_

Round 1

Reviewer 1 Report

Comments and Suggestions for Authors

This paper introduces a framework leveraging Graph Neural Networks (GNN) to analyze human-object interactions, aiming to recognize human activities more effectively in daily living. The framework automatically encodes correlations among various interactions detected from collected relational data, inferring the presence of different activities and their corresponding environmental objects. Evaluated using the Toyota Smart Home dataset, the proposed framework demonstrates significantly improved activity detection performance compared to conventional feed-forward neural networks, achieving a classification accuracy of 0.88 for different daily activities. Moreover, the integration of encoded relational data information enhances object detection performance, increasing accuracy from 0.71 to 0.77 compared to GNNs without joint prediction. However, there are some problems in this paper that suggest improvement

1. The references introduced in the related work, it is recommended to supplement and replace some of the more representative and authoritative research results in the last three years, on the basis of which the existing methods are cited to be improved, thus illustrating that the ideas in this paper are frontier.

2. Pay attention to the order in which the pictures appear, for example, the citation of Figure 1 appears first in the text, and then the picture of Figure 1 appears.

3. In 209 line, Figure 3 whether the citation is wrong, should be Figure 2.

4. When all the mathematical symbols appearing in the formulae appear for the first time, it is recommended that the meaning expressed by the mathematical symbols be explained, to make it easier for the reader to read and understand.

5. In the experimental part, there are fewer comparative methods. It is suggested to add representative methods as well as methods that have performed better in the last three years, briefly introduce these methods and add comparative tests.

Author Response

Dear Reviewer,

We would like to extend our sincere gratitude for your constructive feedback and helpful comments on improving our manuscript. We have uploaded a file containing the details of the modifications made.

We hope that these responses adequately address your questions.

Thank you very much for your assistance!

Reviewer 2 Report

Comments and Suggestions for Authors

The paper proposes an approach to identification of human activities based on neural netowrk-based computer vision technologies.

The authors achieve the goal set but the overall presentation of the material requires improvements.

Line 43: collectedimage -> collected image
Line 115: what do "nodes" represent? Must be objects in the analyzed images|videos as described in the related work, but this is not obvious.
Line 157: reference to Fig. 2 appears far later (even after the reference to Fig. 3).
Line 218: why not to list all 8 activities here? Currently, here you have a couple of examples (though a reader would like to see all of them), and list all of them only in line 240 (approx.).
Line 277: what is "object prediction" in the table? What if the purpose of comparing object predision results? Why do not you use a Yolo network (for object detection, or as a baseline for comparison), which often provides very high results in object detection?

Some newer references could be added (currenlty there is only one of 2023 and a few of 2022).

Author Response

(The authors gave the same response as above.)

Reviewer 3 Report

Comments and Suggestions for Authors

The paper introduces a framework to recognise human activities through human-object interactions, leveraging Graph Neural Networks (GNNs) to model these interactions. The authors use GNNs to model interactions between objects and activities and apply the proposed framework to the Toyota Smart Home dataset. Results show superior performance compared to baseline methods in terms of accuracy.

The paper can be improved for the following aspects:

-       The initial reference to assistive robotic applications in the abstract and introduction might be misleading given the paper's focus on human-object interactions. Clarify the relevance of this reference or change it to improve coherence.

-       The paper lacks a comprehensive discussion on the limitations of the proposed framework. Also, incorporating qualitative analysis, such as examples of correctly and incorrectly classified instances, would provide deeper insights into the framework's pros and cons.

-       Include additional performance metrics besides TOP-1 accuracy for a more comprehensive assessment of the framework's effectiveness.

-       Experiments on diverse datasets or under varying conditions should be conducted to demonstrate the framework's robustness and generalizability.

-       Figure 3b is not readable: adjust the size of labels and improve the resolution.

Author Response

(The authors gave the same response as above.)

Round 2

Reviewer 2 Report

Comments and Suggestions for Authors

The authors have carefully addressed the comments. The paper can now be published.

Reviewer 3 Report

Comments and Suggestions for Authors

The authors improved the paper based on the reviewer's feedback.